# Recent Advances in Graphitic Carbon Nitride Based Electro-Catalysts for CO_2_ Reduction Reactions

**DOI:** 10.3390/molecules28083292

**Published:** 2023-04-07

**Authors:** Xinyi Mao, Ruitang Guo, Quhan Chen, Huiwen Zhu, Hongzhe Li, Zijun Yan, Zeyu Guo, Tao Wu

**Affiliations:** 1College of Energy and Mechanical Engineering, Shanghai University of Electric Power, Shanghai 200090, China; 2Department of Chemical and Environmental Engineering, University of Nottingham Ningbo China, Ningbo 315100, China; 3Municipal Key Laboratory of Clean Energy Technologies of Ningbo, University of Nottingham Ningbo China, Ningbo 315100, China; 4Key Laboratory of Carbonaceous Wastes Processing and Process Intensification of Zhejiang Province, University of Nottingham Ningbo China, Ningbo 315100, China

**Keywords:** graphitic phase carbon nitride (g-C_3_N_4_), electrocatalysis, CO_2_ reduction, single atom catalyst, novel catalyst carrier

## Abstract

The electrocatalytic carbon dioxide reduction reaction is an effective means of combating the greenhouse effect caused by massive carbon dioxide emissions. Carbon nitride in the graphitic phase (g-C_3_N_4_) has excellent chemical stability and unique structural properties that allow it to be widely used in energy and materials fields. However, due to its relatively low electrical conductivity, to date, little effort has been made to summarize the application of g-C_3_N_4_ in the electrocatalytic reduction of CO_2_. This review focuses on the synthesis and functionalization of g-C_3_N_4_ and the recent advances of its application as a catalyst and a catalyst support in the electrocatalytic reduction of CO_2_. The modification of g-C_3_N_4_-based catalysts for enhanced CO_2_ reduction is critically reviewed. In addition, opportunities for future research on g-C_3_N_4_-based catalysts for electrocatalytic CO_2_ reduction are discussed.

## 1. Introduction

Fossil fuels have been used extensively to meet humanity’s primary energy needs since the beginning of the industrial revolution [1,2] and this has led to excessive emissions of carbon dioxide into the air [3], resulting in a high CO_2_ concentration of 420 ppm in the air and consequent global warming [4]. Therefore, in recent years, many countries have started to pay more attention to climate change and have introduced measures to mitigate the emission of carbon dioxide into the atmosphere. Among the possible technologies to reduce CO_2_ emissions into the air, the conversion of CO_2_ into valuable chemicals is considered a viable and promising option [5,6].

To date, a variety of methods have been used to convert CO_2_ [7], including biochemical [8,9], photochemical [10,11], thermochemical [12,13] and electrochemical processes. Electrocatalytic reduction of CO_2_ has many advantages over other CO_2_ conversion technologies [14], such as the following: (1) the reaction can be carried out at room temperature and atmospheric pressure; (2) the reaction can lead to the formation of carbon-based fuels that are conventionally produced based on non-renewable feedstock; and (3) the reduction products can be tuned by adjusting the operating parameters and using different electrocatalysts. These advantages have led to widespread interest in the research on the electrocatalytic reduction of CO_2_. However, the low catalytic activity and poor stability of electrocatalysts are the main obstacles [14] limiting the large-scale conversion of CO_2_. Therefore, there is still a need to develop novel catalysts for the electrocatalytic reduction of CO_2_ that are efficient and durable.

The g-C_3_N_4_ is a chemically stable polymeric semiconductor composed of different isomers of C_3_N_4_, which was first synthesized by Melon in 1834 [15]. There are generally five isomers of C_3_N_4_, namely pseudocubic-C_3_N_4_ (p-C_3_N_4_), cubic-C_3_N_4_ (c-C_3_N_4_), α-C_3_N_4_, β-C_3_N_4_, and graphite-C_3_N_4_. The first four phases all have an atomic density close to that of diamond and are excellent thermal conductors. The g-C_3_N_4_ has the smallest relative forbidden band and is the most stable isomer of carbon nitride.

The synthesis of g-C_3_N_4_ usually involves heating inexpensive nitrogen-rich precursors such as cyanamide, dicyanamide, melamine, thiourea, and urea to remove the amino group. The residue of such a process, a light-yellow powder, is the g-C_3_N_4_. In general, the g-C_3_N_4_ is thermally stable at temperatures up to 600 °C, insoluble in organic solvents, and very stable in acids and bases [16,17,18]. The excellent chemical and physical stability of g-C_3_N_4_ makes it a good choice as a catalyst support in heterogeneous catalysis.

Current research on the application of g-C_3_N_4_ in heterogeneous catalysis(Figure 1) mainly includes photocatalytic water splitting [6,19,20], photocatalytic carbon dioxide reduction [21,22], pollutant degradation [23,24], photocatalytic ammonia synthesis [25,26], photoelectrochemical catalysis [27,28], electrocatalytic hydrogen evolution [29,30], electrocatalytic oxygen evolution [31,32], electrocatalytic water splitting [33,34], electrocatalytic carbon dioxide reduction, electrocatalytic oxygen reduction [35,36], electrocatalytic organic oxidation [18,37], photothermal hydrogenation of carbon dioxide [38,39], photothermal catalytic hydrogen evolution [40,41], etc. The g-C_3_N_4_ has shown a good capability in these catalytic reactions.

Due to the high nitrogen content of g-C_3_N_4_, the Lewis sites and Bronsted sites of the g-C_3_N_4_ structure enhance the adsorption of CO_2_ [42]. In addition, the carbon in g-C_3_N_4_ has a high affinity for oxygen-binding intermediates (_*_OCH_x_, O and OH) in the reactions that results in the production of deep reduction products such as methane (CH_4_) [43]. This property increases the efficiency of the electrocatalytic reduction of CO_2_ and leads to the production of more valuable hydrocarbon products [44]. Bulk g-C_3_N_4_, as a big sheet of lamellar structure, has a low specific surface area (<10 m^2^g^−1^) [45] when used as an electrocatalyst [46] resulting in low electrical conductivity [47] and poor catalytic effect [48] which subsequently limits its use as a catalyst support due to the lack of exposed active sites. Therefore, it is necessary to increase the electrical conductivity, specific surface area, and number of exposed active sites of g-C_3_N_4_ before it can be used as an excellent electrocatalyst and catalyst support [49].

This review focuses on the recent advances in the electrocatalytic reduction of CO_2_ using g-C_3_N_4_ as a catalyst and/or catalyst support. The properties and role of the g-C_3_N_4_ in the electrocatalytic reduction of CO_2_ are discussed. The synthesis, functionalization, and modification of g-C_3_N_4_ materials as electrocatalysts for enhanced CO_2_ reduction are summarized. Finally, the future opportunities for g-C_3_N_4_ as a catalyst for electrocatalytic CO_2_ reduction are discussed.

### 1.1. Fundamentals of Electrocatalytic Reduction of CO_2_

Carbon dioxide is a stable molecule. The carbon dioxide reduction reaction (CO_2_RR) is a commonly used approach to convert CO_2_ into valuable chemicals and is classified as a heterogeneous reaction [6,50]. One-carbon products of CO_2_RR include carbon monoxide (CO), methane (CH_4_), formaldehyde (CH_2_O), methanol (CH_3_OH), and formic acid (HCOOH), while two-carbon reduction products include ethylene (C_2_H_4_), ethane (C_2_H_6_), acetylene (C_2_H_2_), ethanol (C_2_H_6_O), and oxalic acid (H_2_C_2_O_4_), among others. The heterogeneous catalytic conversion of CO_2_ at the electrode surface consists of three main steps: (i) the electrode material loses its absolute linearity due to the adsorption of the normal linear CO_2_ molecule; (ii) the conversion of the C-O bond to a C-H bond by proton-coupled electron transfer; and (iii) the intermediate is desorbed from the catalytic electrode and then diffuses into the electrolyte [51].

Since CO_2_ is structurally stable and its carbon-oxygen double bond is energetic and not easily broken, from a thermodynamic point of view, an equilibrium potential must be applied during the CO_2_RR reaction to reduce CO_2_, as shown in Table 1. This is the reversible hydrogen electrode potential applied for the conversion of carbon dioxide into various products by electrocatalysis. However, the actual electrode potential required for the reduction reaction is more negative than the equilibrium potential [14,52]. This is because in the actual reaction, the carbon dioxide molecule is adsorbed onto the catalyst and gains a single electron, leading to the formation of the key intermediate CO_2_^−^. In this step, the reaction requires a significant amount of energy to convert the linear carbon dioxide molecule into a bent radical anion, making the actual potential more negative than the standard potential.

In the CO_2_RR reaction, different reduction products are formed via different reaction pathways, which can be divided into four categories depending on the reaction intermediates, namely, formaldehyde, carbene, glyoxal, and enol-like intermediates.

Mono-carbon products are usually formed via the formaldehyde pathway (Figure 2a), which involves the synergistic transfer of protons and electrons to CO_2_ to form a carboxylic intermediate (COOH^−^). The intermediate then accepts electrons and hydrogen ions to form HCHO. The carbene pathway (Figure 2b) is the most likely reaction pathway when CO_2_ is bound to the active site via its carbon atom, which gains electrons and H^+^ and then loses hydroxide to form a carbon monoxide intermediate, which is converted to carbon monoxide (*CO^−^). By gaining electrons and H^+^ again, it loses hydroxyl to form carbene intermediates (CH_2_); this is the carbene pathway which is the most likely pathway. Glyoxal and enol-like pathways are common in the formation of multi-carbon (C2–C3) products [53]. The glyoxal pathway (Figure 2c) involves the formation of a coordination bond between the two oxygen atoms of carbon dioxide and a catalytic active site [54,55,56]. The pathway for glyoxal is the formation of a double O coordination bond between carbon dioxide and the catalytic active site, loss of the hydroxyl group to give the formyl radical *HCO after continuous addition of electrons and H^+^, and then dimerization to form the glyoxal intermediate (C_2_H_2_O_2_). C-C bond coupling via C1 or C2 intermediates leads to an aldehyde-free enol intermediate, but there is much uncertainty about the key intermediates involved in C-C bond coupling.

### 1.2. Unique Properties of g-C_3_N_4_ as an Electrocatalyst

Surface catalysis and high mass transfer efficiency are the characteristics and advantages of electrocatalytic reactions. The g-C_3_N_4_ is a layered material with a huge specific surface area and large number of active sites. It has other advantages, such as easy adjustment of the electron distribution at the interface, high acid-base stability, etc. All these advantages suggest that g-C_3_N_4_ can be used as an electrocatalyst for various applications.

#### 1.2.1. Morphology of g-C_3_N_4_

The morphology of g-C_3_N_4_ varies depending on the preparation method and precursors. The nitrogen-rich precursor is thermally polymerized directly to a massive solid g-C_3_N_4_ with a low specific surface area and porosity. The g-C_3_N_4_ with high specific surface area and large porosity (up to 830 m^2^g^−1^ and 1.25 cm^3^g^−1^) [57] was obtained by precursor treatment [58]. In terms of ion/carrier transport, g-C_3_N_4_ has a distinct advantage. When used in multiphase catalysis, porous g-C_3_N_4_ can enhance electrocatalysis by providing solution, electrolyte, and gas channels on the exposed surface, thus improving ion transfer and diffusion processes between different substances. Different preparation methods lead to g-C_3_N_4_ materials with different properties, making them useful as catalytic materials and/or supports for various applications. g-C_3_N_4_ with a large surface area and uniform pore size can be prepared by using the template method [57], and by modulating the temperature and pyrolysis time [59,60,61]; the specific surface area, pore size, and pore volume of g-C_3_N_4_ have been varied, allowing the morphology of the catalyst to be adjusted (Table 2) [62]. The specific surface area and pore size of g-C_3_N_4_ obtained by thermal polymerization of different precursors at the same temperature of 550 °C and in an air atmosphere for 2 h are also different [59,60,63,64].

#### 1.2.2. Surface Active Sites

Catalytic activity is usually related to the active sites where reactants and intermediates are adsorbed and charge transfer takes place [62]. The pyridinic N atom on the heptazine (Figure 3 has a strong electron accepting capacity, making the g-C_3_N_4_ surface an active site for initiating electrochemical reactions and enhancing the electrocatalytic properties of g-C_3_N_4_ [65]. The surface reactivity varies depending on the position of the surface atoms [66]. For the construction of an integrated composite catalyst system, the many active sites of g-C_3_N_4_ (with floating bonds at the pore edge) may also allow better dispersion and binding with auxiliary catalysts or other coupling materials. In the field of mesoporous or macroporous g-C_3_N_4_ catalyst heterogeneous catalysis, its large specific surface area and porosity provide a high density of active sites [49,67]. For the eCO_2_RR, g-C_3_N_4_ provides plenty of N-sites, which enhances the material’s ability to bind CO_2_ and increases the local concentration of CO_2_ around the catalyst [68]. When g-C_3_N_4_ is used as a catalyst, the state of the active site can be changed by adjusting the nitrogen or carbon elements in the structure [44]. When g-C_3_N_4_ is used as a support, the active site of the g-C_3_N_4_ involved in the catalytic reaction changes due to different loading conditions. g-C_3_N_4_ forms coordination bonds with doped elements as the active site of the catalyst [69,70].

#### 1.2.3. Stability

The g-C_3_N_4_ is normally produced by pyrolysis of nitrogen-containing precursors such as urea, dicyandiamide, and melamine at 500 °C, resulting in exceptional thermal stability. Thermal stability begins to decrease when the temperature exceeds 600 °C. Thermal decomposition begins at 650 °C and is completely degraded at 700 °C [71]. g-C_3_N_4_ is not afraid of strong acids and bases; in the preparation of g-C_3_N_4_ by template method, the template can be removed by washing with strong acid [72,73], and its chemical properties are stable [65,73], so g-C_3_N_4_ is widely used as a catalyst carrier. Its stable chemical properties make it a promising organic framework for a single atom catalyst. When g-C_3_N_4_ is used as a catalyst support, its numerous porous channels and active sites can reduce the transfer distance between the catalyst and the electrolyte or solution, accelerate the reaction rate, and protect the catalyst from corrosion for a short period of time.

## 2. Graphitic Carbon Nitride-Based Catalysts for Electrocatalytic CO_2_RR

The g-C_3_N_4_-based catalysts are divided into three main categories: pristine g-C_3_N_4_, metal doped g-C_3_N_4_, and non-metal doping g-C_3_N_4_. Their applications in the field of electrocatalytic carbon dioxide reduction are reviewed, and the synthesis methods of these catalysts as well as the product selectivity control of g-C_3_N_4_-based catalysts for electrocatalytic carbon dioxide reduction reactions are described in detail.

### 2.1. Pristine g-C_3_N_4_

The g-C_3_N_4_ with a specific surface area of 10 m^2^g^−1^ was obtained by holding dicyandiamide in a covered crucible that was heated to 550 °C at a heating rate of 3 °C per minute and kept isothermal for 4 h. The Faraday efficiency (FE) of pristine g-C_3_N_4_ electrocatalysis of carbon dioxide to carbon monoxide in a 0.1 M KHCO_3_ electrolyte is approximately 5% [70,74,75,76]. Compared to the g-C_3_N_4_ nanosheet molecules (with a surface area of 235 m^2^g^−1^) [77] and bulk g-C_3_N_4_ (with a surface area of 8 m^2^g^−1^) [49] and others obtained by conventional methods, the ultra-thin polarized g-C_3_N_4_ layer (2D-pg-C_3_N_4_), which was obtained by hydrothermal stripping (thickness: ~1 nm), has a larger specific surface area of 292.4 m^2^g^−1^, which subsequently enhances the adsorption effect of CO_2_. The overall foam-like structure promotes the diffusion of CO_2_ molecules to the active surface of the catalyst. The ultrathin layered structure of 2D-pg-C_3_N_4_ enables the faster release of electrons from the polarized Melem subunit, which promotes the CO_2_ reduction reaction. At a potential of 1.1 V vs. Ag/AgCl, CO_2_RR achieved a total Faraday efficiency of 91%, resulting in the conversion of CO_2_ to CO (80%) and formic acid (11%), almost completely blocking the HER process. At a potential of −1.2 V, the current density of 2D-pg-C_3_N_4_ reached 3.05 mA cm^−2^, almost 30 times that of bulk g-C_3_N_4_. The production of CO also increased by 17.1 times.

Compared with g-C_3_N_4_ with a complete crystal structure, which can only reduce CO_2_ to CO [44,49,76,78], the π-electron leaving domains of the engineered vacancies in the g-C_3_N_4_ conjugated skeleton and the effect of the vacancies on g-C_3_N_4_ in electrocatalytic CO_2_ reduction reactions have been investigated by Density Flooding Theory (DFT) calculations. A g-C_3_N_4_ obtained by thermal stripping is close to the theoretically calculated N vacancy (vacancy engineered) and is named DCN. A suitable N vacancy can change the geometry of g-C_3_N_4_ and adjust its adsorption strength for key intermediates, which not only increases the desorption energy barrier of *CO intermediates but also limits the formation of CO and H_2_. This reduces the activation energy barrier of CO_2_ reduction to CH_4_ and promotes CH_4_ formation. The carbon atom in DCN was identified as the active site of the CO_2_RR reaction according to DFT calculations. The presence of the N-vacancy changes the tri-coordinating carbon atom around the N vacancy to a di-coordinating carbon atom, making the carbon atom more unsaturated and more prone to combine with the *CO intermediate to form a stronger bond. This makes the desorption of the *CO intermediates more difficult, preventing the formation of CO and allowing further conversion to CH_4_ [79]. The nitrogen-rich DCN electrocatalyst showed high activity in the electrocatalytic reduction of CO_2_ over the whole potential range (Figure 4b). At a potential of −1.27 V vs. RHE, the Faraday efficiency of CH_4_ can reach 44% (Figure 4a), and the current density of CH_4_ generation reaches 14.8 mAcm^−2^, which are 6.3 and 7 times more effective than the ordinary bulk g-C_3_N_4_ catalysts, respectively, when reacting in CO_2_-saturated 0.5 M KHCO_3_, indicating that nitrogen vacancies in carbon nitride can enhance the electrocatalytic reaction.

The modification effect of g-C_3_N_4_ can be achieved by a special treatment of g-C_3_N_4_ itself or the precursor compound, which changes the morphology and electronic structure of g-C_3_N_4_ and increases the electrical conductivity and selectivity of the catalyst for certain products. Table 3 shows the electrocatalytic performance of the pristine g-C_3_N_4_ catalysts for the electrocatalytic CO_2_ reduction reaction. The use of the modified g-C_3_N_4_ as a catalyst support can further increase the efficiency of the electrocatalytic CO_2_ reduction reaction.

### 2.2. Metal Doped g-C_3_N_4_

g-C_3_N_4_ is an excellent catalyst support. At present, various applications of metal doped g-C_3_N_4_ catalysts have been studied in many fields, but less so in CO_2_RR. The metal doped g-C_3_N_4_ catalysts generally have higher electrical conductivity and catalytic activity than non-metal doping g-C_3_N_4_ catalysts due to the electronic properties of the metal atoms. The metal catalysts were classified into single metal doped g-C_3_N_4_ catalysts, bimetallic-doped g-C_3_N_4_ catalysts, and ternary composite catalysts.

#### 2.2.1. Single Metal Doped g-C_3_N_4_

Single metal doped catalysts are currently one of the most used types of electrocatalysts for the CO_2_ reduction reaction. The metal doped catalysts discussed in this article are mainly nanocatalysts. Monometallic catalysts are classified as monatomic catalysts, metal cluster catalysts, and metal nanocatalysts depending on the size of the metal particles.

The active site has the greatest influence on the CO_2_RR activity and selectivity of metal catalysts [80]. Single atom catalysts are catalysts in which metals are uniformly and individually loaded as single atoms onto metals, metal oxides, two-dimensional materials, and molecular sieves, using the single atom as the catalytic active center for the catalytic reaction. Studies have integrated and summarized the stabilization mechanism of single atom catalysis on different supports, where MN_x_, MS_x_, and other stable structures (M is monatomic) with heteroatoms (N, S, P, etc.) are generated [81] on carbon supports. The electronic properties and catalytic performance of a single atom depend on its coordination with the nitrogen and sulfur atoms in the support. The potential advantages of the single atom catalyst on g-C_3_N_4_ support are as follows: (1) the porous structure of g-C_3_N_4_ has a large specific surface area, which can improve the loading rate of metal atoms and create more active sites [82]; (2) the delocalization of the π-electron in the conjugated framework of g-C_3_N_4_ can change the electronic and the catalytic properties of the monatomic center [44]; (3) strictly single atom layers can facilitate the adsorption and diffusion of reactant molecules from either sides of g-C_3_N_4_ on separate single atoms; (4) g-C_3_N_4_ is a good model catalyst, which make it easier to identify uniform active sites and predict catalytic performance using chemical theoretical methods; (5) single-atom anchoring can promote or activate the original catalytic activity of two-dimensional materials [83].

Studies have shown that transition metal atoms (Fe, Ti, Ru, V, Cr, Ir, Mn, Co, Ni, Cu, Rh, Sc, Pd, Au, Ag, Pt) can be combined with a single layer of g-C_3_N_4_ to form a single atom catalyst using first-principles calculations [84,85]. Low temperature embedding Cr and Mn in g-C_3_N_4_ allows the preparation of promising single atom catalysts. Three single dispersed transition metal atoms (Fe, Co, Ni) have been modified in the structure of g-C_3_N_4_. In all optimized structures, the transition metal is located in the hexagonal hole of g-C_3_N_4_ and forms strong coordination bonds with the adjacent mono-atom. Figure 5a–d shows the structural optimization of single atom catalysts of g-C_3_N_4_ and three transition metals. The strength of the M-N bond formed by metal atoms and g-C_3_N_4_ varies with the difference between the atomic radius and valence electron number of cobalt and iron. Stable M-C_3_N_4_ structures are formed by strong M-N bonds [36]. The adsorption process of CO_2_ on the catalyst was studied by DFT calculations. It was found that CO_2_ exhibits weak physical adsorption on Ni-C_3_N_4_ and that the adsorption configuration of CO_2_ on Co-C_3_N_4_ and Fe-C_3_N_4_ is chemical adsorption. In addition to the M-C bond, an M-O bond was also formed between the CO_2_ molecules and the metal atoms of the catalyst. During adsorption, large number of electrons are transferred from Co or Fe atoms to CO_2_ via M-C and M-O bonds. The density of states (DOS) analysis explains that the different adsorption configurations are caused by the different distribution of the d orbitals after the doping of the transition metal atoms. The differences in the adsorption conditions lead directly to the differences in the initial hydrogenation products. *CO_2_ + H^+^ + e^−^ → *COOH is the most favorable first protonation step on Ni-C_3_N_4_, and *CO_2_ + H^+^ + e^−^ → *OCHO is the first step of protonation on Co-C_3_N_4_ and Fe-C_3_N_4_. The carbon dioxide reduction reaction on the surface of three catalysts was studied by constructing a thermodynamic reaction network. The results show that the three catalysts can inhibit the formation of H_2_, CO, and HCOOH, and the end products tend to be CH_3_OH and CH_4_. The main rate-determining steps of Ni-C_3_N_4_ and Fe-C_3_N_4_ in the process of CH_3_OH and CH_4_ formation are the steps to form the intermediate *CHO, so their selectivity is limited. However, the rate-determining steps of the process on Co-C_3_N_4_ are different and show a high methanol selectivity and the lowest initial reaction potential (UL = −0.65 V, as shown in Figure 5e). Therefore, the single atom catalyst based on g-C_3_N_4_ is advantageous to achieve deep CO_2_ reduction at a low electrode potential.

To investigate the effect of the catalysts Co-C_3_N_4_, Fe-C_3_N_4_ and Mg-C_3_N_4_ on the weak CO hybridization (Figure 5f), the reaction pathways of these catalysts were investigated. As shown in Figure 5g, the analytical energy barrier of Mg-C_3_N_4_ (0.13 eV) is much lower than that of Fe-C_3_N_4_ (1.37 eV) and Co-C_3_N_4_ (1.52 eV) [86]. Mg atoms are thought to desorb CO more readily than Fe and Co atoms. CO temperature program desorption (TPD) and in-situ attenuated total reflection infrared (ATR-IR) spectroscopy was used to demonstrate the CO desorption capability of Mg-C_3_N_4_. In contrast to the large CO desorption peaks on Fe-C_3_N_4_ and Co-C_3_N_4_, Mg-C_3_N_4_ shows no CO desorption peak, indicating that CO can be easily desorbed from the Mg sites (Figure 5h). Similarly, significant changes in current density were observed on Fe-C_3_N_4_ and Co-C_3_N_4_ during the electro-response tests in Ar and CO atmospheres. The larger differences in the current density of Fe-C_3_N_4_ (0.16 mAcm^−2^) and Co-C_3_N_4_ (0.09 mAcm^−2^) between the cases with CO and with Ar (Figure 5i) indicate the stronger CO adsorption capacity of the Fe and Co sites compared to the Mg sites (0.006 mAcm^−2^). No significant *CO was found on Mg-C_3_N_4_, although CO was produced in large quantities, indicating that the produced CO was well desorbed (Figure 5i,k). In contrast, the presence of distinct *CO bands on the Fe and Co-C_3_N_4_ electrodes confirmed the significant adsorption of CO on Fe and CO sites (Figure 5l). The results of the TPD, the electro-response measurement, and the ATR-IR spectra show that CO exhibits weak desorption at the Mg site, which is in good agreement with the theoretical calculation results. The electrocatalytic CO_2_ reduction reaction takes place in an H-cell electrolysis cell with a turnover frequency (TOF) of 18,000 h^−1^ for the Mg-C_3_N_4_ catalyst and the Faraday efficiency of CO in the KHCO_3_ electrolyte reaches ≥ 90%. The electrochemical reduction of CO_2_ in the flow cell can achieve a current density of −300 mAcm^−2^ and ensure a Faraday efficiency of 90%. The results show that the metal in the s-block can be used for highly efficient electrochemical reduction of CO_2_ to produce CO.

Unlike monoatomic catalysts, which focus on the dispersed state of individual metal atoms, monometallic nanoparticle catalysts exhibit a much more diverse state of metal element presence. It has been demonstrated experimentally and through theoretical calculations that the interaction of Au with g-C_3_N_4_ causes the Au surface to become extremely electron-rich, thereby facilitating the adsorption of the key chemical intermediate *COOH [87]. Similarly improved CO_2_RR performance was observed on Ag NPs loaded with g-C_3_N_4_ (Ag/C_3_N_4_), which also had electron-rich Ag surfaces [48]. The Ag_2_O precursor was incompletely decomposed under hydrothermal conditions to form super-stable oxides and nanosilver and loaded onto g-C_3_N_4_, and the super-stable oxides in the catalyst improved the binding energy of the *COOH intermediate [48]. The rate-determining step of the electrocatalytic CO_2_ reduction reaction is changed from electron transfer to proton transfer due to the strong interaction of the Ag nanoparticles (NPs) with the g-C_3_N_4_ support in the catalyst via the Ag-N bond. This improves the performance of electrocatalytic CO_2_ reduction, and the Faraday efficiency of CO at −0.7 V vs. RHE can reach 94% at low potential. Ag-decorated B-doped g-C_3_N_4_ catalysts were synthesized by loading Ag NPs on boron-doped g-C_3_N_4_, and the electrochemical reduction properties of the catalysts were investigated by theoretical calculations and experiments [70]. The DFT calculations show that the Ag-B-g-C_3_N_4_ catalyst can significantly reduce the adsorption free energy for the formation of the *COOH intermediate. In addition, the electron accumulation at the Ag-B-g-C_3_N_4_ interface can promote electron transit and increase the electrical conductivity. The simulation results show that the addition of B atoms and Ag NPs can significantly improve the eCO_2_RR performance of g-C_3_N_4_. The electrocatalysts g-C_3_N_4_, B-g-C_3_N_4_, and Ag-B-g-C_3_N_4_ were prepared for comparative experiments and it was proved by XPS, XRD, TEM, and other characterization methods that CO can only be generated by Ag-B-g-C_3_N_4_, proving that Ag is the only active site. Electrochemical impedance spectroscopy (EIS) analysis showed that the Ag atom has a catalytic effect on electron transport. An Ag-B-g-C_3_N_4_ catalyst with an average diameter of 4.95 nm has a total current density of 2.08 mAcm^2^ and the Faraday efficiency of CO is 93.2% at a potential of −0.8 V vs. RHE.

A study was carried out by performing extensive DFT calculations on Cu-C_3_N_4_ model catalysts and comparing their CO_2_ reduction potential with Cu(111) surface and standard Cu-NC complexes, showing that Cu-C_3_N_4_ has better CO_2_ reduction activity, lower starting potential, and a significantly higher C2 formation rate compared to Cu-NC (Figure 6a–d) [78].

Cu_2_O/CN was obtained by immobilizing Cu_2_O nanocubes on the structure of g-C_3_N_4_ by chemical precipitation. On the one hand, the g-C_3_N_4_ framework provides an anchor center for the in-situ growth of Cu_2_O, which promotes uniform dispersion of Cu_2_O and exposes more active sites. On the other hand, g-C_3_N_4_ shows good CO_2_ adsorption and activation capabilities. At the interface between the g-C_3_N_4_ support and the Cu_2_O NPs, the CO_2_ is adsorbed onto the g-C_3_N_4_, generating *CO intermediates. The *CO intermediates generated on the g-C_3_N_4_ have the possibility of C-C coupling with the C atoms in the intermediates generated on the Cu_2_O surface, which improves the active site and increases the electrocatalytic kinetic rate, thus increasing the yield of C_2_H_4_ [68]. The specific steps are the reduction of incoming CO_2_ to CO by combining 2H^+^ and 2e^−^ on g-C_3_N_4_. The CO formed on g-C_3_N_4_ can then be transported to the Cu_2_O site due to the stronger bonding between CuO and CO and the increased local CO concentration and residence time near the Cu_2_O surface due to the high surface coverage of *CO. Further reduction of CO or CO dimers at the active site leads to the formation of C_2_H_4_, followed by C_2_H_4_ desorption from Cu_2_O/CN. The key step in the transfer of CO from C_3_N_4_ to Cu_2_O was shown by calculations to be due to the synergistic effect of g-C_3_N_4_ and Cu_2_O. At −1.1 V vs. RHE, the Faraday efficiency of the Cu_2_O/CN composite on C_2_H_4_ is 32.2% and the local current density is −4.3 mAcm^−2^. At −1.1 V vs. RHE for at least 4 h, the Cu_2_O/CN catalyst maintained its stable performance and structure.

Compared to pure CuO nanosheets and spherical CuO particles, g-C_3_N_4_ plays a role in increasing the specific surface area and exposing active sites in the CuO/g-C_3_N_4_ catalyst, providing new opportunities for CO_2_ adsorption and promoting mass transfer kinetics. In addition, the interaction of pyridine-N and copper oxide contributes to C-C coupling, further enhancing the activity of the CO_2_ reduction reaction, with Faraday efficiencies of up to 64.7% for all C2 products below −1.0 V vs. RHE [89]. MnO_2_/g-C_3_N_4_ [90] and ZnO/g-C_3_N_4_ [90] catalysts can be prepared by a simple pyrolysis method. Possible catalytic routes for the reduction of CO_2_ to formates in alkaline media with additional bases such as triethylamine are shown in (Figure 6e). First, CO_2_ molecules can combine with water molecules on the surface of g-C_3_N_4_ to form carbonic acid. CO_2_ acts as a Lewis acid and is activated by triethylamine to form amphoteric carbamate intermediates. The amphoteric intermediates formed on the metal surface and the active metal oxides on the other side play a role in the activation, adsorption, and dissociation of hydride molecules. The hydride molecule is transported from the catalyst surface to the active intermediate, the amphoteric carbamate ion (Schiff base), to form carbamate, which undergoes acid–base neutralization in the aqueous medium and is then converted to formic acid. When graphite carbon nitride donates electrons, the electrons reach the surface of metal oxide and help the hydride (-H) dissociate from the surface of the metal oxide, giving it easy access to the electron-deficient carbamate intermediate to form formic acid. For formates, the MnO_2_/g-C_3_N_4_ catalyst has a Faraday efficiency of 65.28%, while the ZnO/g-C_3_N_4_ catalyst has a Faraday efficiency of 80.99%, which is related to the properties of the metal itself.

Monoatomic metal catalysts are popular with researchers due to their excellent electrochemical properties and have been well studied theoretically, but actual experiments are still rare due to the difficulty of their synthesis. Nanocluster catalysts have a very high surface area and unique surface structural features. The doping of the mono-metal on g-C_3_N_4_ increases the active sites for the reaction and improves the stability of the nanometallic particles in the catalyst and the selectivity of the products. Table 4 shows the electrocatalytic parameters of the single metal doped g-C_3_N_4_ catalysts in the electrocatalytic carbon dioxide reduction reaction.

#### 2.2.2. Bimetallic Doped g-C_3_N_4_

Bimetallic catalysts often outperform monometallic catalysts of the same metal composition due to the synergistic interaction between the different atoms in the bimetallic catalyst [93]. The dynamic structure and chemical changes on the surface of the bimetallic catalyst during inhomogeneous catalytic reactions make the synergistic mechanism more complex [94]. Studying the catalytic reaction process and reaction pathway of bimetallic catalysts and selecting suitable metal elements for g-C_3_N_4_ modification can improve stability and catalytic activity.

Cu_x_Ru_y_CN was obtained by modifying copper–ruthenium bimetallic compounds on the surface of π-conjugated g-C_3_N_4_. The Cu_x_Ru_y_CN samples exhibited excellent BET surface area, pore size, and pore volume due to the appropriate Cu and Ru doping ratio, which attracted reaction molecules and provided active sites for enhanced electrocatalytic processes. The Mott–Schottky effect is caused by the formation of metal-semiconductor interfaces between Ru, Cu, and g-C_3_N_4_ heterojunctions, which significantly increases the efficiency of charge separation and prevents reverse flow from the metal to the semiconductor. The mixed state of CuO and Cu_2_O acts as an active center for the adsorption and activation of CO_2_, while RuO_2_ acts as a center for the enrichment of holes for the synergistic transfer of H protons to promote the reduction of CO_2_. In an air or Ar atmosphere, the current density of the reaction decreases to below 0.05 mAcm^−2^ when the applied potential is −1.5 V, indicating that the high current density of the Cu_x_Ru_y_CN catalyst in the reaction is related to the flow of CO_2_ and its reduction. Moreover, the current density of Cu_x_Ru_y_CN remains constant at an applied potential of −1.4 V vs. Ag/AgCl for at least 2000 s, indicating its high stability in the CO_2_ reduction process [95].

The CuSe/g-C_3_N_4_ catalyst can be obtained by anchoring hexagonal CuSe nanoplates on g-C_3_N_4_ nanosheets by hydrothermal method [96,97]. The morphologies of the hexagonal CuSe nanoplates before and after anchoring are shown in SEM images (Figure 7a,b). The internal electric field formed between the electron coupling Cu and Se on the electrode is confirmed by the DFT calculation, and the electrons move from the g-C_3_N_4_ nanosheets to the CuSe nanoplates. The results show that CuSe is the active site of the catalyst and that CO_2_ is activated on the surface of the CuSe nanoplates and controlled by the activation process. The CuSe/g-C_3_N_4_ with 50% CuSe nanoplate content was tested and showed the best catalytic performance. At −1.2 V vs. RHE, its CO Faraday efficiency was 85.28% (Figure 7c,d). The Faraday efficiency is 1.47 times higher than that of pure CuSe nanoplates, which is due to the addition of g-C_3_N_4_ nanosheets with a planar structure, which provide a larger specific surface area.

The catalyst g-C_3_N_4_/Cu_2_O-FeO was obtained by dissolving iron salt, copper salt, and g-C_3_N_4_ in a certain ratio in triethylene glycol and reacting in an autoclave for 12 h [98]. High resolution transmission electron microscopy (HRTEM) results showed that the prepared nanocomposites have Cu_2_O-FeO mixed metal oxide nanoclusters uniformly distributed on the surface of the g-C_3_N_4_ nanosheets. The size is about 10 nm (Figure 7e–g). This composite catalyst is used as an electrode for the electrochemical reduction of CO_2_ and exhibits high catalytic activity. The total current density is 4.65 mAcm^−2^, the overpotential is −0.865 V vs. NHE, and the applied potential is −1.60 V vs. Ag/AgCl. The maximum CO Faraday efficiency is 84.4% (Figure 7h,i). The conversion rate is up to 10,300 h^−1^ and the selectivity is 96%. This improvement is the result of the close interfacial interaction between g-C_3_N_4_ and the metal oxide (Cu_2_O-FeO), which leads to a larger electrochemically active surface area and oxygen vacancies on the surface.

The C_3_N_4_/Co(OH)_2_/Cu(OH)_2_ catalyst is a bimetallic hydroxide catalyst whose synthesis is divided into two steps [92]. The first step consists of the synthesis of Co NPs by reduction of Co^2+^ ions on the surface of C_3_N_4_ with the strong reducing agent NaBH_4_. Co^2+^ ions are hydrolyzed in water to form Co(OH)_2_. In the second stages, these Co NPs undergo an electrical exchange process on C_3_N_4_ where the Co NPs are exchanged for Cu atoms to form Cu(OH)_2_ in an aqueous solution. Overall, the synthesis process culminates in the deposition of cobalt and copper hydroxides (C_3_N_4_/(Co(OH)_2_/Cu(OH)_2_) on the surface of C_3_N_4_. By changing the surface morphology during the primary cell replacement process, more active sites and suitable adsorption-matrix interactions can be achieved. The electrocatalytic activity of C_3_N_4_/(Co(OH)_2_/Cu(OH)_2_ is more than three times that of C_3_N_4_/(Co(OH)_2_/Cu(OH)_2_ due to the synergistic effect of cobalt and copper hydroxide.

Bimetallic doped g-C_3_N_4_-based catalysts, with g-C_3_N_4_ material as the catalyst support, are able to stably load bimetals onto the g-C_3_N_4_ structural skeleton. The bimetals not only create interactions with g-C_3_N_4_ similar to those between metal and g-C_3_N_4_ in monometallic catalysts, but also have their unique synergistic effects between the bimetals to further enhance the electrocatalytic effect. Current research in this area is based on nanoparticles. Dual atom catalysts (DACs), obtained by modulating the morphology of metal particles, and exploiting the interaction between the two metals can effectively overcome some of the application limitations of SACs. Table 5 shows the electrocatalytic parameters of the bimetallic doped g-C_3_N_4_-based catalyst in eCO_2_RR.

#### 2.2.3. Ternary Compound Catalyst

A ternary complex catalyst usually consists of three parts. Metal atoms or metal clusters are the key components of the ternary complex catalyst that control the catalytic performance or catalytic activity of the catalyst. g-C_3_N_4_ provides abundant active sites for ternary complex catalysts. It can also enhance the dispersion and interaction with co-catalysts or other coupling materials for the construction of an integrated composite catalyst system. Graphene, CNT, porous carbon, molybdenum disulfide, and other compounds in the terpolymer catalyst are mainly co-catalysts, which mainly help to increase the specific surface area of the catalyst, create a larger active site, and improve the conductivity.

Mn-C_3_N_4_/CNT is a monatomic catalyst with Mn-N3 as the active site. This conclusion was reached by analyzing the N1s XPS spectra of Mn-C_3_N_4_/CNT and C_3_N_4_/CNT (Figure 8a,b). The results showed that the N atom of C-N-C was the coordination site for the formation of the Mn-NX structure [99]. The quantitative EXAFS curve fit analysis (Figure 8c) was used to calculate the structural parameters of Mn-C_3_N_4_/CNT. The Mn-NX structure had a coordination number x of about 3.2, indicating that an isolated Mn atom was coordinated cubically by the N atom and the final coordination structure was Mn-N_3_. The CO_2_ adsorption, activation, and transformation processes on Mn-C_3_N_4_/CNT were investigated using in situ X-ray absorption spectroscopy and DFT calculations. The three N atoms in the Mn central coordination reduce the free energy barrier for CO_2_ to form important intermediates (Figure 8d). At a low overpotential of 0.44V, the Mn-C_3_N_4_/CNT catalyst showed a CO Faraday efficiency of 98.8% in 0.5M KHCO_3_ solution, and the CO partial current density was 14.0 mAcm^−2^. The addition of CNT mainly improved the catalyst conductivity and the catalytic effect of CO_2_RR.

The ternary compound catalyst CoPPc@g-C_3_N_4_-CNTs was obtained by polymerizing cobalt phthalocyanine (CoPc) on three-dimensional (3D) g-C_3_N_4_ nanosheets and carbon nanotubes. The results of the electrocatalytic experiment showed that the CO Faraday efficiency is 95 ± 1.8% at −0.8 V vs. RHE and the conversion frequency is 4.9 ± 0.2 s^−1^, indicating good long-term stability within 24 h. The hydrothermal synthesis (Figure 8e) of the protonated g-C_3_N_4_ nanosheets and CNTs improved the immobilization uniformity at high catalyst loading and the interaction between the molecular catalyst and the conductive support compared to similar hybrid electrocatalysts prepared by drop-drying or dip-coating. The electrochemically active surface area was increased, the structure was improved, and the active sites were enriched, resulting in excellent catalytic performance [101].

The Ag-S-C_3_N_4_/CNT [69] ternary composite catalyst showed exceptional performance in eCO_2_RR with a high current density of 21.3 mAcm^−2^ at −0.77 V vs. RHE. The highest CO Faraday efficiency in the H-cell is over 90% (Figure 8f,g). When the same catalyst is applied in the flow cell configuration, the current density is shown to exceed 200 mAcm^−2^, which is essentially the current density required for industrial CO_2_RR. The Faraday efficiency of CO is greater than 80% over a wide range of potentials. DFT and electrochemical methods were used to further investigate the catalytic mechanism of the nanocomposites (Figure 8h). The synergistic effect of Ag NPs, sulfur elements, the C_3_N_4_ framework, and carbon nanotube supports results in very efficient performance of the eCO_2_RR. The outermost catalytic surface consists of Ag NPs, sulfur atoms, and C_3_N_4_ frameworks, on which CO_2_ is directly converted to CO. As a result, electron accumulation at the interface of Ag-S-C_3_N_4_/CNT and S-C_3_N_4_/CNT is combined with the excellent charge transport performance of CNT and the properties of the S-material C_3_N_4_ to improve the electrical properties of Ag-S-C_3_N_4_/CNT nanocomposites. The Faraday efficiency of the ternary electrocatalyst Au-CDots-C_3_N_4_ [99] doped with precious metal is as high as 79.8% at a potential of −0.5V, and the current density increases by a fact of 2.8 at −1.0 V (where the Au loading is only 4%). DFT calculations and experimental observations have shown that the synergistic effect of Au NPs, CDots, and C_3_N_4_ and the adsorption capacity of CDots for H^+^ and CO_2_ are the sources of the high activity of Au NPs in CO_2_RR. At the same time, the combination of CDots and Au-C_3_N_4_ can accelerate the rate of charge transfer in the reaction. The charge transfer process in CO_2_ reduction was studied using EIS of Au-CDots-C_3_N_4_ (Figure 8i). It was found that the radius of CDot-C_3_N_4_ was obviously smaller compared to C_3_N_4_, and Au-CDot-C_3_N_4_ was smaller compared to Au-C_3_N_4_, which was attributed to the enhanced conductivity due to the high charge transfer ability of the CDots. Therefore, the CDots in the Au-CDots-C_3_N_4_ terpolymer have good adsorption capacity for CO_2_ and H^+^ and play a leading role in promoting the formation of CO.

The Cu-g-C_3_N_4_/MoS_2_ [102] ternary composite catalyst of g-C_3_N_4_, MoS_2_ and copper nanoparticles (Cu NPs) showed good electrocatalytic activity in eCO_2_RR, and the Faraday efficiencies for methanol and ethanol were 19.7% and 4.8%, respectively. Compared with Cu-g-C_3_N_4_ and Cu-MoS_2_, the EIS results of Cu-g-C_3_N_4_/MoS_2_ composites (Figure 8j) show that the interaction between MoS_2_ and g-C_3_N_4_ enhances the electron and charge transfer on the catalyst surface. The Cu-g-C_3_N_4_/MoS_2_ composites have the lowest resistivity, as indicated by the smallest semicircle radius. The EIS results show that compared to Cu-g-C_3_N_4_ and Cu-MoS_2_, the charge transfer in the Cu-g-C_3_N_4_/MoS_2_ composite is improved after the combination of g-C_3_N_4_ and MoS_2_, which makes a greater contribution to the electrocatalytic activity in CO_2_ reduction. Cu-g-C_3_N_4_/MoS_2_ composites have lower ohmic resistance, and thus have better catalytic activity in CO_2_ reduction than Cu-g-C_3_N_4_ and Cu-MoS_2_.

The common feature of the terpolymer catalyst is that the catalyst can achieve a high current density in CO_2_RR. The addition of a co-catalyst mainly plays a role in improving the conductivity and enlarging the active site, while the combination of different metals and different co-catalysts produces various synergistic effects to achieve the catalytic improvement effect. Table 6 shows the electrocatalytic parameters of the ternary composite catalyst for eCO_2_RR.

### 2.3. Non-Metal Doping g-C_3_N_4_

Among the current doping techniques, the insertion of heteroatoms can change the electrical structure of g-C_3_N_4_. Doping of g-C_3_N_4_ with metals and non-metals is the most common type of elemental doping. Non-metallic elements enter the g-C_3_N_4_ system more easily than metallic components. For example, the elements O, C, S, N, and F are doped into the g-C_3_N_4_ system by replacing the elements C, N, and H in the heptazine structural unit. The non-metallic doping strategy is to improve the catalytic performance of the catalysts by increasing the adsorption of carbon dioxide and the selectivity of products.

Figure 9a,b shows the Gibbs free energy conversion diagrams for the reduction of CO_2_ to carbon monoxide by g-C_3_N_4_ and B-g-C_3_N_4_, respectively. From this, it can be seen that doping with elemental boron reduces the free energy barrier for the reaction to produce CO and improves the eCO_2_RR performance of the catalyst. As shown in Figure 9c, the charge transfer resistance of B-g-C_3_N_4_ is much lower than that of g-C_3_N_4_. The B atom can effectively enhance the electron transport of g-C_3_N_4_ [70]. A similar conclusion is drawn from Figure 8h, where sulfur doping lowers the Gibbs free energy barrier of CO conversion [69]. The enhanced intrinsic electrical properties and CO_2_ reactivity of the Ag NPs, elemental sulfur, the g-C_3_N_4_ framework, and CNT support synergistically promote electron transfer and stabilize the reaction intermediates. Figure 9d shows that the g-C_3_N_4_/CNT doped with sulfur elements has a higher current density and better electrochemical properties at the same electrode potential compared to the g-C_3_N_4_/CNT undoped with any element.

g-C_3_N_4_/MWCNT [74] composites can be prepared by the typical thermal polymerization method of multi-walled carbon nanotubes (MWCNTs) and g-C_3_N_4_ [49]. The addition of multi-walled carbon nanotubes improved the ability of the catalyst to conduct electricity, as well as its total specific surface area and the number of active sites. Analysis of the XPS spectrum of C1s (Figure 9e) showed that the C1 peak at 284.7 eV was classified as a sp2 carbon–carbon double bond, the C1 peak at 288.2 eV as the N=C-N group of the triazine ring in g-C_3_N_4_, and the C1 peak at 285.9 eV as a C-OH species in MWCNT. Crucially, the additional C1 peak at 287.5 eV is the sp3 C-N covalent bond formed in the g-C_3_N_4_/MWCNT composite. This bond shows that g-C_3_N_4_ is not bound to MWCNTs by simple physical adsorption. This is because g-C_3_N_4_ is co-synthesized into the graphite network of MWCNTs through C-N covalent bonds. The active site of the composite is precisely the C-N covalent bond formed between g-C_3_N_4_ and MWCNTs, which can selectively reduce CO_2_ to CO. The maximum Faraday efficiency of carbon monoxide reaches 60% at a potential of −0.75 V vs. RHE. Table 7 shows the electrocatalytic parameters of non-metal doping g-C_3_N_4_ catalyst for CO_2_RR.

At present, there is little research on non-metal doping of g-C_3_N_4_ in the field of electrocatalytic carbon dioxide reduction reaction. The pristine g-C_3_N_4_ has a low conductivity. Compared with the metal doped g-C_3_N_4_ catalyst, the non-metallic doping can only slightly improve the conductivity of the g-C_3_N_4_ catalyst. In addition, the non-metallic doping can form a new C-X coordination bond (X is a non-metallic element) on the surface of g-C_3_N_4_, increasing the active site of the catalyst and improving the catalytic effect of the catalyst.

## 3. Method for the Synthesis of g-C_3_N_4_-Based Catalysts

The g-C_3_N_4_-based catalysts are prepared by thermal polycondensation [104], thermal decomposition method [105], hydrothermal synthesis [48,70,87], and reduction method [86], which are described in detail below.

### 3.1. Thermal Polycondensation

Thermal polycondensation [106,107] is a polycondensation reaction of nitrogen-containing triazine ring structure precursors at 400~600 °C. In a specific atmosphere (air, nitrogen, argon, or hydrogen), in a muffle or tube furnace, at a heating rate of 2 to 10 degrees Celsius per minute, in a crucible with a lid to maintain a specific temperature for 2 to 4 h, this method allows for bulk access to g-C_3_N_4_ [49].

The doping strategy based on this method is to obtain precursor powders by direct mixing and milling of non-metallic or metallic compounds with nitrogen-containing organic compounds, or by reacting aqueous solutions of non-metallic compounds with nitrogen-rich organic compounds and obtaining precursor powders by heating or freeze-drying. The precursor powders obtained from the above steps are heated under a specific atmosphere to obtain non-metal doped g-C_3_N_4_-based catalysts [69], single metal or bimetallic doped g-C_3_N_4_-based catalysts [78,86,88,95,103] wherein the precursor powders obtained after mixing the metal salt solution and urea and freeze-drying under an inert atmosphere are prepared by thermal polycondensation to obtain metal oxide-doped g-C_3_N_4_-based catalysts [90]. Examples of non-metal doping g-C_3_N_4_-based catalysts are as follows: S-C_3_N_4_ containing sulfur is obtained by thermal polymerization of thiourea as a precursor [69]. Boron-doped g-C_3_N_4_ is obtained by direct mixing and milling of boric acid and urea followed by thermal polycondensation [70] or by mixing and dissolving a phosphoric acid solution with urea and freeze-drying the solvent to obtain a bulk precursor [108]. Phosphorus-containing g-C_3_N_4_ can be obtained by mixing of phosphoric acid and urea followed by thermal polycondensation [109].

### 3.2. Thermal Decomposition Method

The thermal decomposition process essentially involves the pyrolysis of selected feedstocks at a specific temperature and under a specific atmosphere (N_2_, NH_3_, Ar, or H_2_). The pyrolysis temperature is between 200 °C and 500 °C, depending on the decomposition temperature of the metal salts. The precursor is usually a mixture of carbon skeleton and metal complexes or a precursor containing a sacrificial template. The thermal decomposition method is usually used for the preparation of single atom catalysts, where g-C_3_N_4_ is used as a support and metal compounds are mixed with it and decomposed by heating under a specific atmosphere to obtain a single atom catalyst with metal monomers anchored to g-C_3_N_4_ [86,100].

### 3.3. Hydrothermal Synthesis

The hydrothermal synthesis method is specified by using a high-pressure reactor as the reaction vessel, selecting a suitable solvent and nitrogen-containing reactants (usually ethanol and dicyandiamide), and controlling the reaction by adjusting the reaction temperature (120–200 °C) and pressure and finally obtaining the g-C_3_N_4_ catalyst. The precursor solutions were mixed with metal salts by adding sodium hydroxide, and the precursor solutions were obtained as g-C_3_N_4_-based catalysts doped with metal oxides or hydroxides in an autoclave by an alkali-assisted synthesis method [92].

The template method is an advanced method of synthesis using thermal solvents. The addition of various templating agents changes the structure and morphology of the g-C_3_N_4_ material. Finally, the compounds used as templates in the catalysts were removed with acid to obtain g-C_3_N_4_-based catalysts with high porosity and high specific surface area (up to 830 m^2^g^−1^ and 1.25 cm^3^g^−1^) [48,70,87]. This method gives good control of the carbon and nitrogen content of the product.

### 3.4. Wet Chemical Reduction

The first step in the wet chemical reduction method is to mix the metal salt solution with g-C_3_N_4_, and the second step is to reduce the metal ions to monoatomic metal uniformly charged on g-C_3_N_4_. There are two methods of reducing the metal ions. In one method, the metal ions uniformly distributed in the pores of g-C_3_N_4_ are reduced by dropwise addition of a reducing agent (sodium borohydride, ethylene glycol, etc.) to obtain a monoatomic catalyst; in the other method, the precursor solution is stirred under a hydrogen atmosphere for 4–10 h and the resulting product is collected by centrifugation and washed several times before being annealed under an argon atmosphere. The most important point in the liquid-phase reduction method is that the experiment must be strictly controlled to avoid monoatomic agglomeration [86].

In the preparation of g-C_3_N_4_-based bimetallic catalysts, the reduction method can be divided into co-reduction, replacement, and sequential reduction methods. Bimetallic nanoparticles with an alloy structure are prepared by the co-reduction method. The g-C_3_N_4_ is thoroughly mixed with a metal salt solution and then reduced together with a reducing agent [96]. Dissolve 100 mg g-C_3_N_4_ in 300 mL DI water, stir for 1 h at room temperature, and add 0.59 g sodium citrate and 7.9 mg silver nitrate to the water solution. Slowly add 20 mL of 0.1 M sodium borohydride dropwise to the solution and stir for 8 h, then filter by centrifugation or filtration. The product is purified with DI water and dried overnight in an oven at 60–80 °C [69]. In the replacement method, a metal is first loaded onto the g-C_3_N_4_ framework and then part of the metal is oxidized by another metal with a higher reduction to obtain a g-C_3_N_4_-based bimetallic catalyst [92]. This uses a sequential reduction method in which one metal is first loaded onto the g-C_3_N_4_ framework and then the other metal is reduced to the original single metal catalyst by the addition of a reducing agent. Unlike the replacement method, the sequential reduction method does not consume the metal originally deposited on the g-C_3_N_4_ framework.

There are suitable synthesis methods for different materials. The advantages and shortcomings of the four synthesis methods are described in Table 8.

## 4. Regulation of Reactant Selectivity by g-C_3_N_4_-Based Catalyst

With g-C_3_N_4_-based catalysts, electrocatalytic carbon dioxide reduction reactions mainly produce hydrogen, methane, carbon monoxide, formic acid, and ethylene. The g-C_3_N_4_-based catalysts modulate the selectivity of the reactants mainly by modulating the active sites on the catalysts, and the different energy barriers for adsorption and desorption of the main reaction intermediates result in higher or lower selectivity of the products. In the case of g-C_3_N_4_-based catalysts doped with Cu metal elements, deeper reactions often occur, producing a variety of two-carbon products such as ethylene, ethanol, and acetic acid. This is due to the properties of copper itself resulting in product selectivity [68,78]. Therefore, the selectivity of the catalyst for the reaction can be modulated by elemental doping and by changing the surface structure. Elemental doping, which is described in detail in the section on catalyst preparation, can be used to modulate the active site in two ways. One is that the element forms a new coordination bond with the carbon or nitrogen in the g-C_3_N_4_ material, creating a new active site that affects the selectivity of the reaction. The other is that the anchoring of the element in the active site of the g-C_3_N_4_ material affects the selectivity of the reaction due to the unique properties of the element itself.

Modification of the surface structure involves the modulation of the morphology and structure of the g-C_3_N_4_-based catalyst itself. Modulation of g-C_3_N_4_ materials is usually done using the template method. In the hard template method, the g-C_3_N_4_ precursor is filled with an inorganic templating agent with microscopic pore structure, thermally condensed in situ, and then the hard template is removed with a solvent such as hydrofluoric acid to obtain the modulated g-C_3_N_4_ catalyst. In the soft template method, a surfactant, ionic liquid, or amphiphilic block polymer is used as a template to condense with the precursor compound of g-C_3_N_4_, and then a thermal polycondensation reaction is carried out to obtain a porous g-C_3_N_4_ with the soft template removed. For g-C_3_N_4_-based catalysts doped with metal atoms, the size of the metal particles can be varied to adjust the selectivity. For metal nanocatalysts, catalysts with metal cluster structures and metal nanocatalysts all have different effects on the reaction [110]. The reaction selectivity of the catalyst can be modified by adjusting the morphology and structure [111]. For bimetallic doped g-C_3_N_4_-based catalysts, the selectivity of the product can be adjusted by adjusting the composition ratio of the two metals.

## 5. Summary

g-C_3_N_4_-based catalysts have a wide range of promising applications in multiphase catalytic reactions, such as photocatalytic degradation, photo/electrocatalytic water splitting, and photo/electrocatalytic carbon dioxide reduction. Moreover, their unique electronic structure, abundant active sites, and high stability make them well-suited for use as electrocatalysts. In this paper, the synthesis of g-C_3_N_4_-based catalysts is summarized and g-C_3_N_4_-based catalysts are classified into pristine g-C_3_N_4_, metal doped g-C_3_N_4_, and non-metal doping g-C_3_N_4_. The practical applications of g-C_3_N_4_-based catalysts in CO_2_RR under different doping modes are discussed. In addition, the role of different types of g-C_3_N_4_-based catalysts in modulating reaction selectivity and synthetic ideas are discussed.

While g-C_3_N_4_ catalysts can be obtained by simple thermal polycondensation, single atom doped g-C_3_N_4_-based catalysts are difficult to prepare and most studies on single atom doped g-C_3_N_4_-based catalysts are still at the stage of theoretical calculations and the experimental part has not been fully explored. Some monometallic nanoparticle catalysts, which are close to the monoatomic catalyst structure, also have very high electron conversion efficiencies and are currently the most studied CO_2_RR electrocatalysts, with the disadvantage that the current density is low and the conversion effect of non-precious metals is not as excellent as that of precious metals. For bimetallic doped g-C_3_N_4_-based catalysts, the interaction between the internal bimetal and g-C_3_N_4_ is similar to that between the metal and g-C_3_N_4_ in monometallic catalysts, and there is a unique synergy between the bimetals that further enhances the electrocatalytic effect and improves the product selectivity and Faraday efficiency of the non-precious metal for CO_2_RR products. Ternary compound catalysts combine the advantages of the previous catalysts and provide not only higher current densities but also greater product selectivity in the reaction, but the synthesis method is complex and most long-term stability tests are limited to 24 h and further long-term stability studies are needed.

## 6. Outlook

It is expected that further technical development of the g-C_3_N_4_-based catalyst will enable large-scale CO_2_ reduction applications [112]. The morphology, atomic composition, crystal surface, and defect type of the g-C_3_N_4_-based catalyst influence the CO_2_ reduction. The g-C_3_N_4_ is a polymeric semiconductor composed of C and N atoms. In coordination designs, N vacancies [113] or other elemental vacancies [114,115,116] can be introduced on the surface of g-C_3_N_4_, changing the surrounding electronic structure and coordination environment to anchor the metal atom [117]. Alternatively, uniform coordination sites can be designed on the surface of g-C_3_N_4_ to adsorb stable metal atoms and metal precursors and prevent their agglomeration and migration, resulting in a monatomic catalyst. Other common atoms or groups that have strong interactions with metal atoms, such as O, S, P, -C≡C-, etc., can also be considered as active components of g-C_3_N_4_-based catalysts. In addition, diatomic catalysts and ternary catalysts, which have higher metal loading, more versatile active sites, and unique active reactions compared to monoatomic catalysts, are also worthy of investigation [118]. At present, research on monoatomic and diatomic catalysts based on g-C_3_N_4_ is still largely at the stage of theoretical calculations, and experimental synthesis and testing has only just begun. The difficulty and challenge in the preparation of such mono- and diatomic catalysts is to exploit the unique chemical and physical properties of g-C_3_N_4_ to make the coordination on g-C_3_N_4_ uniform.

Although there are still significant challenges to overcome, it is widely believed that g-C_3_N_4_-based catalysts have potential in CO_2_RR in the future, especially with advances in synthesis techniques that can translate theory into practical applications.

## Figures and Tables

**Figure 1 molecules-28-03292-f001:**
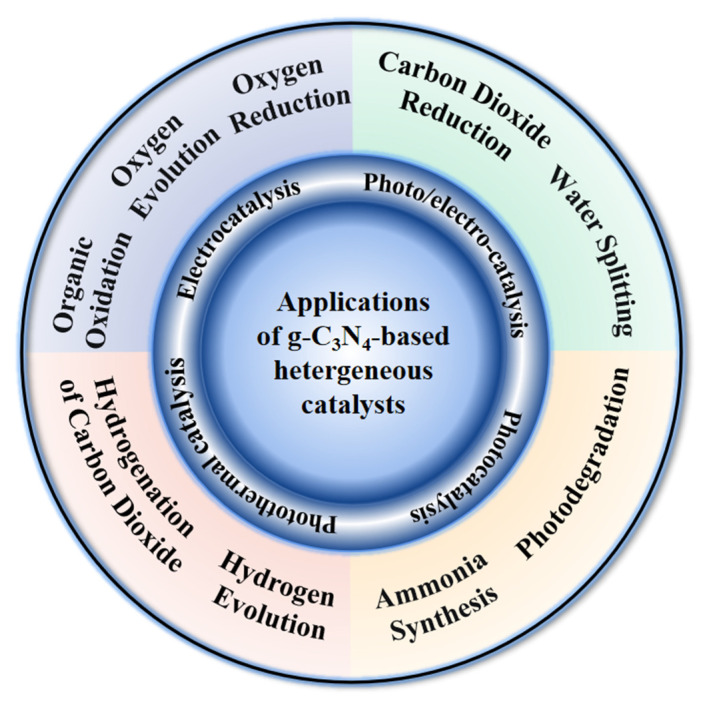
Applications of g-C_3_N_4_ in heterogeneous catalysis.

**Figure 2 molecules-28-03292-f002:**
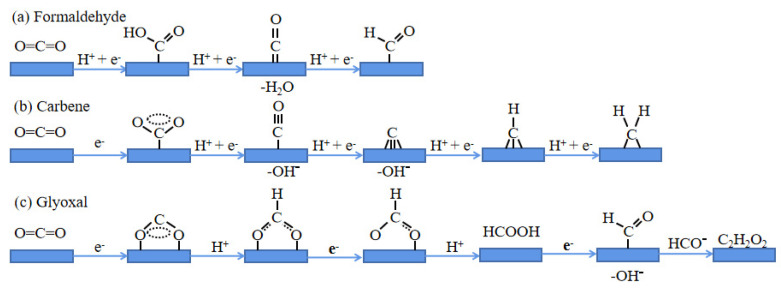
Different ways of reducing carbon dioxide: (**a**) formaldehyde; (**b**) carbene; (**c**) glyoxal [42,54,55,56].

**Figure 3 molecules-28-03292-f003:**
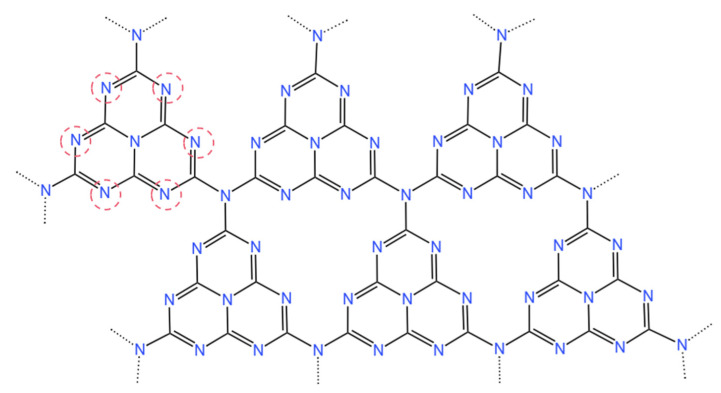
Heptazine structures of g-C_3_N_4_. The nitrogen atom circled by the red dotted line is pyridine nitrogen.

**Figure 4 molecules-28-03292-f004:**
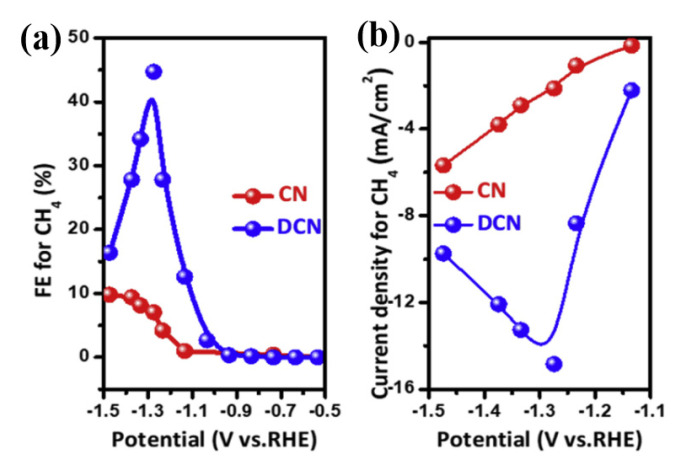
In CO_2_-saturated 0.1 M KHCO_3_, (**a**) CN and DCN produce FE of CH_4_ at each potential. (**b**) Comparison of current densities to produce CH_4_; reprinted with permission from ref. [44], Copyright 2020, Nano Energy.

**Figure 5 molecules-28-03292-f005:**
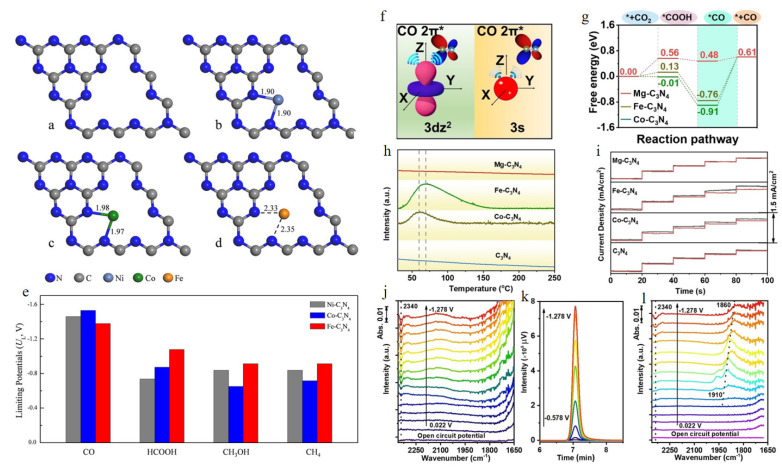
Optimized structure diagram of g-C_3_N_4_ (**a**), Ni-C_3_N_4_ (**b**), Co-C_3_N_4_ (**c**) and Fe-C_3_N_4_ (**d**); (**e**) Histogram of production-related limiting potentials for CO, HCOOH, CH_3_OH, and CH_4_, reprinted with permission from ref. [36], Copyright 2019, ChemSusChem; (**f**) schematic diagram comparison for CO adsorbed on 3 dz^2^ and 3 s orbits; (**g**) free energy diagram of Mg-C_3_N_4_ CO_2_RR, desorption capacity of CO on Mg-C_3_N_4_; (**h**) CO-TPD curve; (**i**) electrode current density in electrical response measurement under Ar and CO; (**j**) Mg-C_3_N_4_ ATR-IR spectra in situ; (**k**) Mg-C_3_N_4_ ATR-IR spectra in situ producing CO gas chromatograph (GC) spectra; (**l**) ATR-IR spectra of Fe-C_3_N_4_ in situ [86], Copyright 2021, Wiley-VCH GmbH.

**Figure 6 molecules-28-03292-f006:**
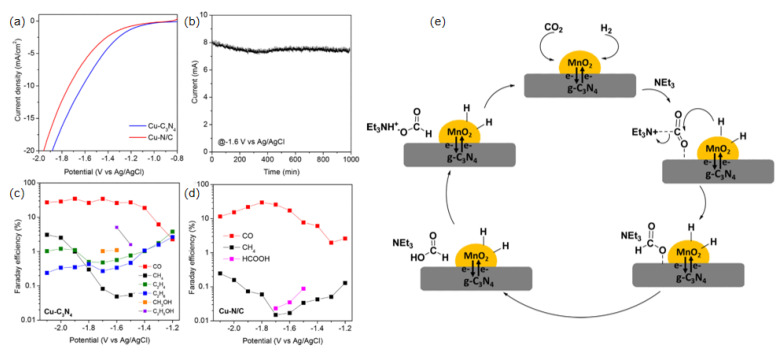
(**a**) CO_2_ reduction polarization curves on two electrocatalysts, measured in CO_2_-saturated 0.1 M KHCO_3_; (**b**) CO_2_ reduction stability test on Cu-C_3_N_4_ electrocatalyst; the Faradaic efficiencies of several products on Cu-C_3_N_4_ (**c**) and Cu-NC (**d**) at different overpotentials, reprinted with permission from ref. [78], Copyright © 2023 American Chemical Society; (**e**) a probable reaction pathway for the hydrogenation of CO_2_ to produce formic acid/formates over a MnO_2_/g-C_3_N_4_ catalyst, reprinted with permission from ref. [88], Copyright © 2023 Elsevier B.V.

**Figure 7 molecules-28-03292-f007:**
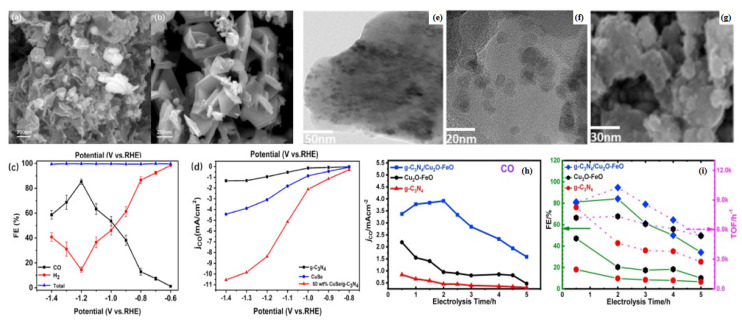
(**a**) 50% CuSe/g-C_3_N_4_ SEM images; (**b**) CuSe nanoplates SEM images; (**c**) Faraday efficiency plots of 50% CuSe/g-C_3_N_4_; (**d**) Partial CO current density of g-C_3_N_4_ nanosheets, CuSe nanoplates, and 50% CuSe/g-C_3_N_4_; reprinted with permission from ref. [96], Copyright © 2023 Elsevier Ltd. HRTEM images of g-C_3_N_4_/Cu_2_O-FeO: (**e**) 50 nm scale; (**f**) 20 nm scale; (**g**) SEM of g-C_3_N_4_/Cu_2_O-FeO at 30 nm scale; (**h**) Current density of CO when using g-C_3_N_4_/Cu_2_O-FeO; (**i**) Curves of TOF and FE of the products with electrolysis time when using g-C_3_N_4_/Cu_2_O-FeO, Cu_2_O-FeO, and g-C_3_N_4_. The olive solid line indicates FE and the magenta dashed line indicates TOF; reprinted with permission from ref. [98], Copyright 2021 Elsevier B.V.

**Figure 8 molecules-28-03292-f008:**
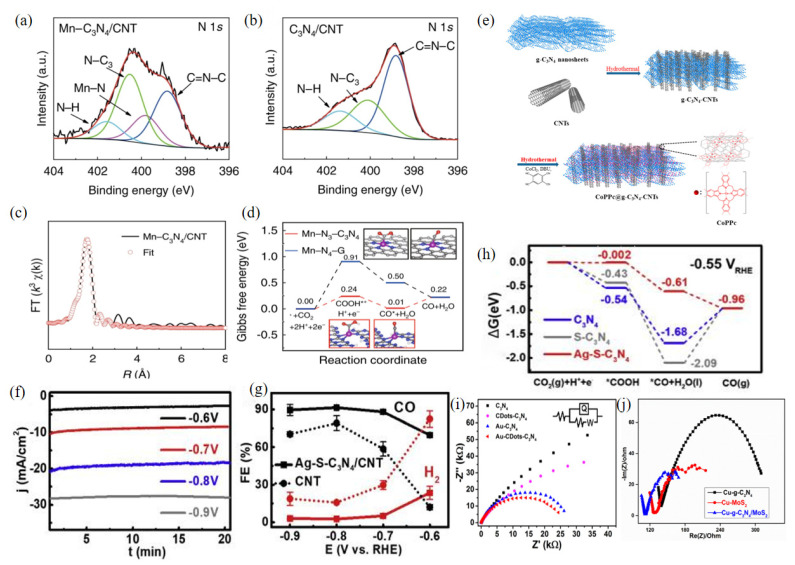
N 1s XPS spectra of Mn-C_3_N_4_/CNT (**a**) and C_3_N_4_/CNT (**b**,**c**) Mn-C_3_N_4_/EXAFS CNT fitting curve in R space. (**d**) The calculated Gibbs free energy diagrams of the electrocatalytic reduction of CO_2_ by Mn-N_3_-C_3_N_4_ and Mn-N4-G; reprinted with permission from ref. [100], Copyright 2020, the author. (**e**) The 3D CoPPc@g-C_3_N_4_-CNT composite synthesis scheme; reprinted with permission from ref. [101], Copyright 2020, American Chemical Society. (**f**) Timing amperograms of Ag-S-C_3_N_4_/CNT at different potentials. (**g**) Comparison of the Faraday efficiency of Ag-S-C_3_N_4_/CNT with bare carbon nanotubes. (**h**) The calculated Gibbs free energy diagram of the electrocatalytic reduction of CO_2_ to CO by Ag-S-C_3_N_4_; reprinted with permission from ref. [69], Copyright 2019 Elsevier Ltd. (**i**) EIS Nyquist plots of 4 wt% Au-CDots-C_3_N_4_ electrode; reprinted with permission from ref. [99], Copyright © 2023, American Chemical Society. (**j**) Cu-g-C_3_N_4_/MoS_2_ electrode EIS measurements; reprinted with permission from ref. [102], Copyright 2022 Elsevier Ltd.

**Figure 9 molecules-28-03292-f009:**
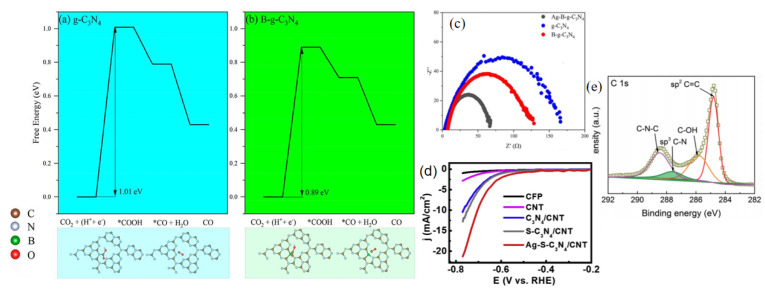
Calculated free energy spectrum of the electrocatalytic reduction of CO_2_ to CO by (**a**) g-C_3_N_4_, (**b**) B-g-C_3_N_4_, (**c**) EIS diagram of g-C_3_N_4_, B-g-C_3_N_4_, and Ag-B-g-C_3_N_4_ catalysts at −0.8 V vs. RHE; reprinted with permission from ref. [70], Copyright 2019 Elsevier Ltd. (**d**) Linear sweep voltammetry curve of g-C_3_N_4_/CNT, (**e**) XPS C 1s spectra of the g-C_3_N_4_/MWCNT composite; reprinted with permission from ref. [74], Copyright 2016 WILEY-VCH Verlag GmbH & Co. KGaA, Weinheim.

**Table 1 molecules-28-03292-t001:** Electrode potentials of selected CO_2_ reduction reaction in a 0.1 M solution of KHCO_3_ at 1.0 atm and 25 °C.

	Chemical Formula and Molecular Formula	Half-Electrochemical Reaction	Potential versus Reversible Hydrogen Electrode (V vs. RHE)
C1	HCOOH 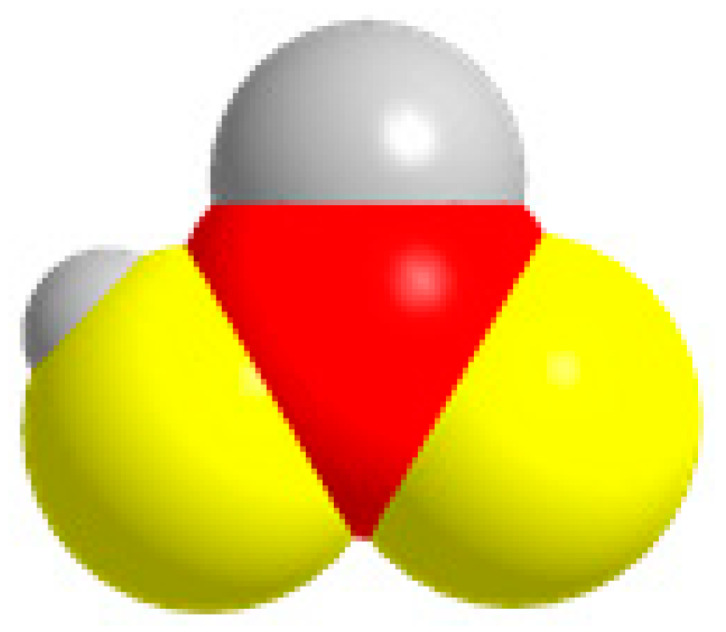	CO_2_ + 2H^+^ + 2e^−^ = HCOOH	−0.651
	CO 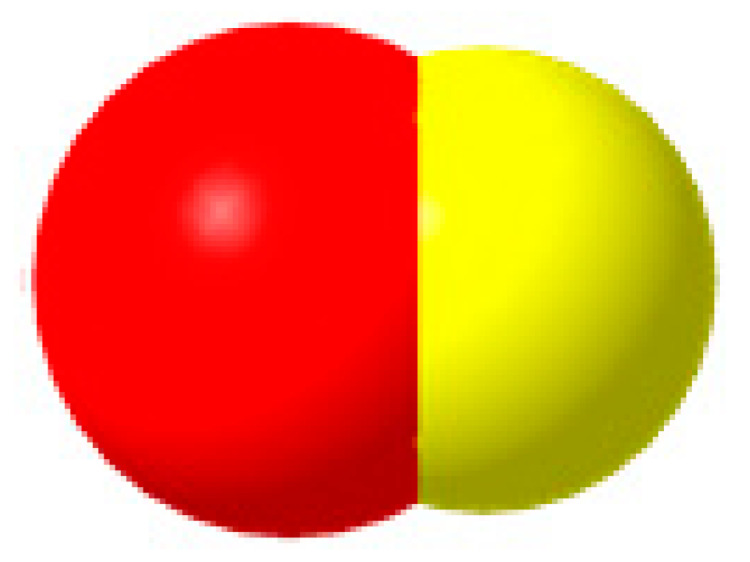	CO_2_ + 2H^+^ + 2e^−^ = CO + H_2_O	−0.507
	CH_2_O 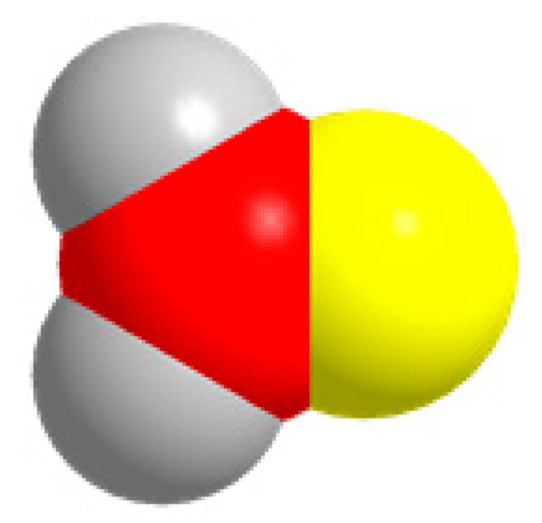	CO_2_ + 4H^+^ + 4e^−^ = CH_2_O + H_2_O	−0.471
	CH_3_OH 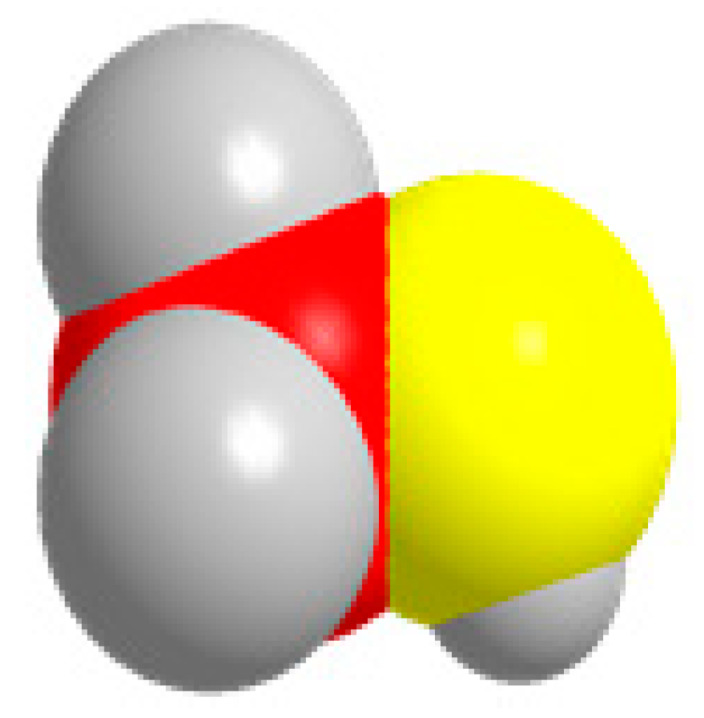	CO_2_ + 6H^+^ + 6e^−^ = CH_3_OH + H_2_O	−0.385
	CH_4_ 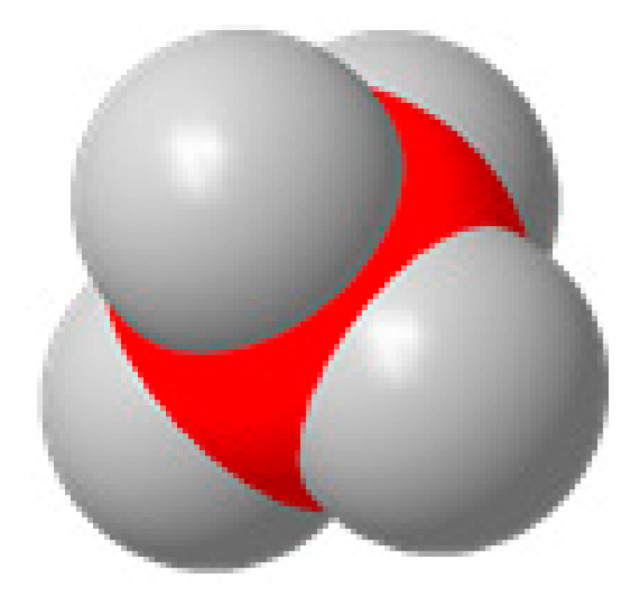	CO_2_ + 8H^+^ + 8e^−^ = CH_4_ + 2H_2_O	−0.232
C2	C_2_H_2_O_4_ 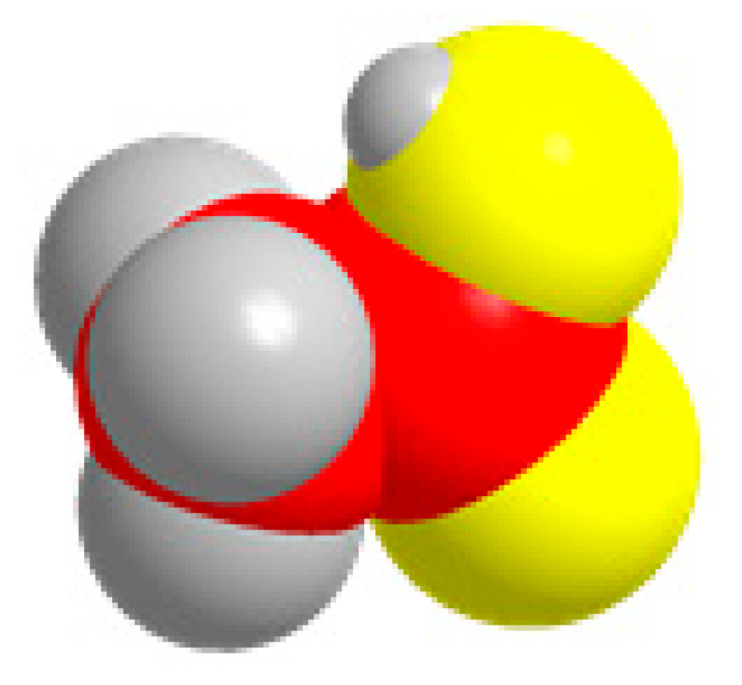	2CO_2_ + 2H^+^ + 2e^−^ = H_2_C_2_O_4_	−0.901
	C_2_H_6_ 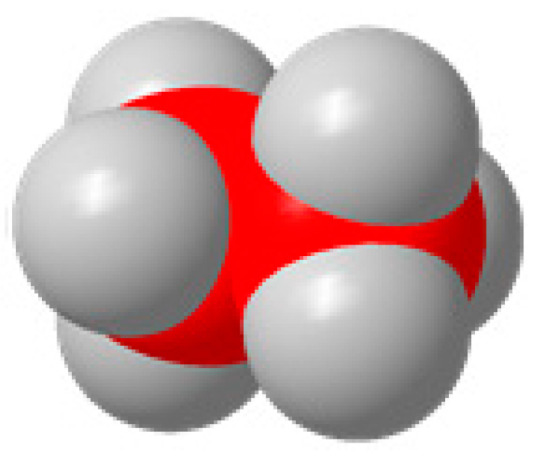	2CO_2_ + 14H^+^ +14e^−^ = C_2_H_6_ + 4H_2_O	−0.261
	C_2_H_4_ 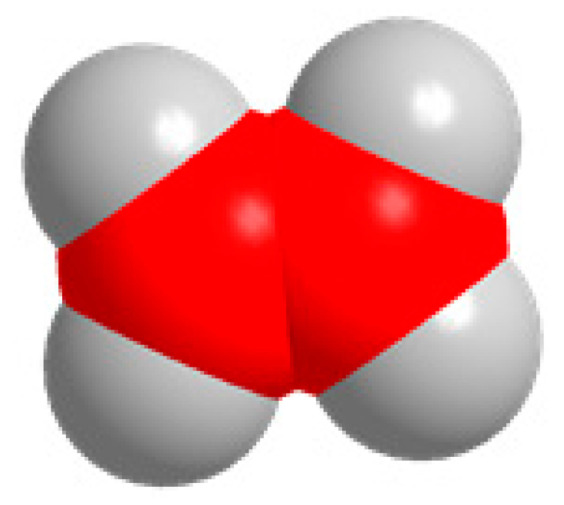	2CO_2_+ 12H^+^ + 12e^−^ = CH_2_CH_2_ + 4H_2_O	−0.337
	C_2_H_5_OH 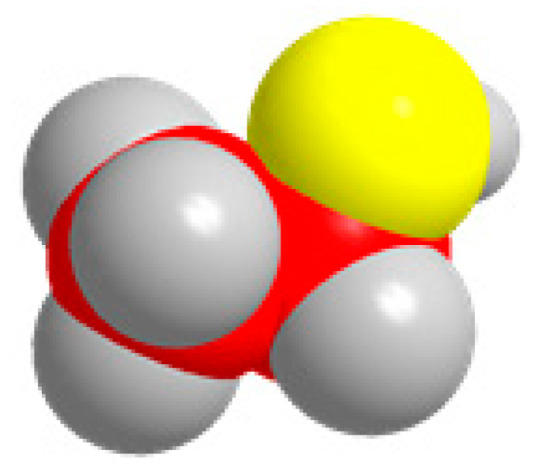	2CO_2_ + 12H^+^ + 12e^−^ = CH_3_CH_2_OH + 3H_2_O	−0.3172

Carbon, oxygen, and hydrogen atoms are red, yellow, and gray, respectively.

**Table 2 molecules-28-03292-t002:** Specific surface area, pore volume, and pore size of g-C_3_N_4_ under different precursor types and different synthesis parameters.

Precursors of g-C_3_N_4_	Reaction Method	Specific Surface Area (m^2^g^−1^)	Pore Volume (mLg^−1^)	Pore Diameter (nm)	Reference
ethylenediamine (EDA) and carbon tetrachloride (CTC)	Hydrothermal synthesis,100 °C	505	0.55	4.2	[57]
EDA and CTC	Hydrothermal synthesis,130 °C	830	1.25	5.1	[57]
EDA and CTC	Hydrothermal synthesis,150 °C	650	0.89	6.4	[57]
Melamine	470 °C, 2 h, air	6.0	0.02	35.2	[59]
Melamine	500 °C, 2 h, air	41.5	0.14	9.2	[59]
Melamine	520 °C, 2 h, air	173.6	0.77	15.6	[59]
Melamine	540 °C, 2 h, air	0.77	0.94	16.5	[59]
urea	550 °C, 0.5 h, air	52	N/A	N/A	[60]
urea	550 °C, 1 h, air	62	0.30	N/A	[60]
urea	550 °C, 2 h, air	75	0.34	N/A	[60]
urea	550 °C, 4 h, air	288	1.41	N/A	[60]
Water-assisted urea	450 °C, 3 h, air	96	0.72	N/A	[61]
Water-assisted urea	450 °C, 5 h, air	106	0.68	N/A	[61]
Dicyandiamide	550 °C, 2 h, air	10	N/A	N/A	[64]
Melamine	550 °C, 2 h, air	8.6	0.02	N/A	[59]
Thiourea	550 °C, 2 h, air	11	N/A	N/A	[64]

N/A indicates that the data is not mentioned in the reference.

**Table 3 molecules-28-03292-t003:** Electrocatalytic parameters of the pristine g-C_3_N_4_ catalysts in eCO_2_RR.

Electrode	Product	FE	Potential (V vs. RHE)	Electrolyte	Current Density (mAcm^−2^)	Ref
Bulk g-C_3_N_4_	CO	5%	−1.2	0.1 M KHCO_3_	ca.0	[70]
g-C_3_N_4_	CO	ca.8%	−1.1	0.1 M KHCO_3_	ca.30	[75]
2D-pg-C_3_N_4_	CO	80%	−0.6	2 M KHCO_3_	3.05	[76]
DCN	CH_4_	44%	−1.27	0.5 M KHCO_3_	14.8	[44]

**Table 4 molecules-28-03292-t004:** Electrocatalytic parameters of single metal doped g-C_3_N_4_ catalysts in eCO_2_RR.

Electrode	Product	FE	Potential (V vs. RHE)	Electrolyte	Current Density (mAcm^−2^)	Ref
Mg-C_3_N_4_	CO	90%	−1.178	KHCO_3_	32	[86]
Ag/g-C_3_N_4_	CO	94%	−0.7	1.0 M KHCO_3_	11.5	[48]
Au/C_3_N_4_	CO	90%	−0.45	0.5 M KHCO_3_	2.56	[87]
Ag/C_3_N_4_	CO	92%	−0.9	0.5 M KHCO_3_	22	[87]
Ag-Decorated B-Doped g-C_3_N_4_	CO	93.20%	−0.8	0.5 M KHCO_3_	2.08	[70]
Fe@C/g-C_3_N_4_	CO	88%	−0.38	0.1 M KHCO_3_	5.5	[91]
ZnO/g-C_3_N_4_	formate	80.99%	−0.934	0.5 M KHCO_3_	ca.33	[90]
Cu_2_O/CN	C_2_H_4_	32.20%	−1.1	0.1 M KHCO_3_	−4.3	[68]
Cu/C_3_N_4_	CO	ca.30%	ca.−1.0	0.1 M KHCO_3_	8	[78]
MnO_2_/g-C_3_N_4_	formate	65.28%	−0.54	0.5 M KHCO_3_	ca.5	[88]
C_3_N_4_/(Co/Co(OH)_2_)	formate	N/A	−0.9	0.5 M KHCO_3_	0.08	[92]

**Table 5 molecules-28-03292-t005:** The electrocatalytic parameters of bimetallic doped g-C_3_N_4_ catalysts in eCO_2_RR.

Electrode	Product	FE	Potential (V vs. RHE)	Electrolyte	Current Density (mAcm^−2^)	Ref
CuSe/g-C_3_N_4_	CO	85.28%	−1.2	0.1 M KHCO_3_	11	[96]
Cu_x_Ru_y_CN	N/A	N/A	−0.8	0.1 M KHCO_3_	0.3	[95]
g-C_3_N_4_/Cu_2_O-FeO	CO	84.40%	−0.9829	0.1 M KCl	3.91	[98]
C_3_N_4_/(Co(OH)_2_/Cu(OH)_2_	formate	N/A	−0.9	0.5 M KHCO_3_	0.23	[92]

**Table 6 molecules-28-03292-t006:** Electrocatalytic parameters of the ternary composite catalysts on eCO_2_RR.

Electrode	Product	FE	Potential (V vs. RHE)	Electrolyte	Current Density (mAcm^−2^)	Ref.
Mn-C_3_N_4_/CNT	CO	98.8%	−0.5	0.5 M KHCO_3_	14	[100]
CoPPc@g C_3_N_4_-CNTs	CO	95%	−0.8	0.5 M KHCO_3_	21.9	[101]
Ag–S–C_3_N_4_/CNT	CO	91.40%	−0.77	0.1 M KHCO_3_	21.3	[69]
NiCu-C_3_N_4_-CNT	CO	ca.90%	−0.8	0.5 M KHCO_3_	ca.14	[103]
NiMn-C_3_N_4_-CNT	CO	ca.90%	−0.8	0.5 M KHCO_3_	ca.12	[103]
Au-CDots-C_3_N_4_	CO	79.80%	−0.5	0.5 M KHCO_3_	0.29	[99]
Cu-g-C_3_N_4_/MoS_2_	CH_3_OH	19.70%	−0.67	0.5 M KHCO_3_	78	[102]

**Table 7 molecules-28-03292-t007:** Electrocatalytic parameters of non-metal doping g-C_3_N_4_ catalysts on eCO_2_RR.

Electrode	Product	FE	Potential (V vs. RHE)	Electrolyte	Current Density (mAcm^−2^)	Ref
S-C_3_N_4_	CO	N/A	−0.77	0.1 M KHCO_3_	N/A	[69]
C_3_N_4_/CNT	CO	N/A	−0.77	0.1 M KHCO_3_	10	[69]
g-C_3_N_4_/MWCNTs	CO	60%	−0.75	0.1 M KHCO_3_	ca. 0.55	[74]

**Table 8 molecules-28-03292-t008:** Advantages and shortcomings of g-C_3_N_4_-based catalyst synthesis methods.

Synthesis Method	Advantages and Disadvantages of Catalyst	The Advantages and Shortcomings of the Method
Thermal polycondensation	Low specific catalyst surface area, high temperature resistance, and good stability	Easy synthesis, high yield, low cost, part of the precursor powder must be uniformly dispersed before participating in the reaction, reaction temperature at (400–600 °C)
Thermal decomposition method	Uniform structure, good heat resistance	Simple reaction process, requires specific atmosphere (Air, N_2_, Ar, H_2_) and temperature requirements (200–500 °C)
Hydrothermal synthesis	Variety of porous catalysts with regular morphology can be produced according to the characteristics of the template, and good heat resistance	Easy to control synthesis, low yields, long preparation cycles, reaction temperatures between (120–200 °C)
Wet chemical reduction	Homogeneous morphology, easy formation of nanocluster structure through doped metal elements, high electrochemical performance	The reaction takes place at room temperature and the reaction steps are cumbersome.

## Data Availability

Not applicable.

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
