# Peer review of "Recent Advances in Graphitic Carbon Nitride Based Electro-Catalysts for CO2 Reduction Reactions"

_molecules, 2023, doi:10.3390/molecules28083292_

Round 1

Reviewer 1 Report

This review presented the recent advances in g-C3N4 toward electrocatalytic carbon dioxide reduction, including the advantages and research prospects of g-C3N4 as a catalyst and catalyst carrier, and discusses in detail the synthesis methods and design strategies of g-C3N4-based materials. It contributes to the development of g-C3N4 in future applications in electrochemistry. This work could be published. However, it needs to undergo revisions beforehand.

The followings are the matters for concern:

(1) In the abstract section, statements are not clearly described.

(2) The application of g-C3N4 materials in electrocatalytic CO2 reduction reactions for generating multi-carbon products could be discussed.

(3) The text formatting of Figure 1 needs to be adjusted.

(4) More literature support is needed for section 1.2.3.

(5) Literature citations must be added under some figures.

(6) Add some keywords that better highlight the paper, e.g. 'novel catalyst carrier' and "synergistic interaction of metals and non-metals"

Author Response

See attachment please.

Reviewer 2 Report

This manuscript overview the reduction of CO2 on the g-C3N4 and the composite catalysts based thereof, which is the hot topic in the field of electocatalysis. However, the quality of the manuscript is unsatisfactory. The manuscript looks like it has been prepared in negligence, without sufficient attention and accuracy.

The text is poorly structured, the narration is intermittent, logical connections between the paragraphs and sections is missing. Text contains a lot of repeats, inaccurate terminology, meaningless or wrongly placed fragments etc. The text is nothing but a compilation of facts without any meaningfull conclusions. Talking about a structure of the g-C3N4 in section 1.2.1, authors have not provided the structural formula even of the repeating unit of this "polymer". The manuscript called "Graphitic Carbon Nitride Based Electrocatalysts for CO2 Reduction Reaction: Syntheses and Applications" does not contain even sub-section devoted to its synthesis. And what "applications" (plural) are meant, if the manuscript is devoted to a single application - CO2 reduction?

Some further examples:

L78 "The bulk material g-C3N4, with a small specific surface area, formed by stacking many layers of g-C3N4 has a small specific surface area" - materal with a small specific surface area has a small specific surface area.

L80 "...has low conductivity and poor catalytic efficiency, when used as an electrocatalyst resulting in low electrical conductivity and poor catalytic property" - low conductivity and poor catalytic efficiency results in electrical conductivity and poor catalytic property (what is "poor catalytic property"?).

L101 "the conversion of the C-O bond to a C-H bond by proton migration or electron transfer" - how can just an electron transfer convert the C-O bond to the C-H bond?

L102 "product is desorbed by conformational transformation" - what does the term "conformational transformation" mean? And how this process can desorb the molecule from the surface?

L157 "...reheat polymerization of the precursor treatment" - this term is meaningless. Polymerization of the treatment?

L161 "Metal-free g-C3N4 materials are considered promising for catalysis." - it should not appear in sections discussing the structure of the material.

L3 "The N atom on the heptane 3 ring..." - the term "heptane ring" is meaningless. g-C3N4 itself does not contain any 7-member rings. The understanding of such terms, anyway, is impossible without the structure drawn.

L36 "The g-C3N4 with BET of 10m2g-1" - BET is a method, not a physical value.

L107 "Single-atom catalysts are single-atom catalysts..." - no comments.

...and that was just a little portion of them.

Graphics quality is poor. For example, proportions of the circle on Fig. 1 are distorted. Permissions are missed for some imported graphics. Text is crowded with typos: indexes without subscript, missing or excessive spaces, missing letters etc. Many abbreviations, lables or specific terms are not described, at least at the first appearance in text.

Considering that the readability, representation quality and correct description of the data are the key points of the review paper, I do not recommend this manuscript for the publication.

Author Response

See attachment please

Round 2

Reviewer 2 Report

The authors adress most of the points. However, there are still significant amount of typos (for example - missing spaces in L90, L98, non-capital letter in Lewis in L333 at P16 etc) and bad wordings (for example what is "hydride molecule", L336 and further). Authors should furhet revise the text quality, aftewards the manuscript may be accepted.

Author Response

Attached is the point-to-point responses to reviewers' comments
